# Process Evaluation of a Participative Organizational Intervention as a Stress Preventive Intervention for Employees in Swedish Primary Health Care

**DOI:** 10.3390/ijerph17197285

**Published:** 2020-10-06

**Authors:** Bozana Arapovic-Johansson, Irene Jensen, Charlotte Wåhlin, Christina Björklund, Lydia Kwak

**Affiliations:** 1Unit of Intervention and Implementation Research for Worker Health, Institute of Environmental Medicine, Karolinska Institute, 17177 Stockholm, Sweden; irene.jensen@ki.se (I.J.); Charlotte.Wahlin@regionostergotland.se (C.W.); christina.bjorklund@ki.se (C.B.); lydia.kwak@ki.se (L.K.); 2Unit of Clinical Medicine, Occupational and Environmental Medicine, Department of Health, Medicine and Caring Sciences, Linköping University, 58183 Linköping, Sweden

**Keywords:** process evaluation, organizational level intervention, implementation, Consolidated Framework for Implementation Research, mixed method design, primary health care

## Abstract

This study is a process evaluation of a trial examining the effects of an organizational intervention (Productivity Measurement and Enhancement System or ProMES) on employee stress. The aims were to explore the implementation process and fidelity to the intervention guidelines, examine the influence of contextual factors (hindrances and facilitators) and explore participants’ experience of working with ProMES. We used the UK Medical Research Council (MRC) guidance to guide the process evaluation. The recruitment, reach and dose delivered were satisfactory and participation high. The employees felt ProMES clarified priorities, gave control and increased participation in decision-making. However, difficulty in obtaining statistical productivity data from the central administration office (a central feature of the intervention) hindered full implementation and regular feedback meetings. Staffing shortages interfered with the implementation process, while having seven design teams and one consultant prevented all occupational groups from working simultaneously. A detailed examination of access to necessary organizational data should be undertaken before implementing ProMES. We recommend a better introduction for new employees, more work on design and packaging and giving employees more training in how to use the software program. The study contributes to our understanding of process evaluations in research into organizational stress management interventions.

## 1. Introduction

Organizational interventions are important for the primary prevention of stress and stress-related mental ill-health because they target known environmental risk factors for mental ill-health such as high job demands, low job control and low social support [1]. However, to date, there is mixed research evidence regarding the effectiveness of organizational-level interventions aimed at the prevention of stress and mental ill-health. The interventions that include a change of work schedules—i.e., shorter or interrupted work schedules [2] have strong support, while more research is needed regarding other types of organizational interventions. A new Cochrane Systematic Review on the effectiveness of organizational level interventions for the reduction (and prevention) of occupational stress in health care workers is underway [3]. This updated review will replace earlier reviews and hopefully shed more light on the state of the art. Furthermore, in order to better understand the limited evidence for the effectiveness of organizational interventions, efficacy and effectiveness studies need to be complemented by comprehensive process evaluations which include the evaluation of process variables [4].

In contrast to effectiveness studies that focus on evaluating the effectiveness of interventions, process evaluations provide information about the implementation process, including what is delivered under what circumstances, participants’ interactions with the delivered activities and the impact of external factors on the delivery [5]. Process variables assessed during process evaluations may be factors within the organizational context or the intervention itself that either hinder or facilitate the implementation process [6]. This information is necessary if policy and practice stakeholders are to understand whether, when and how to use a certain intervention in a specific context. While process evaluations are not uncommon in public health research, the evaluation of organizational interventions which aim to reduce exposure to stress is still predominantly efficacy-oriented and pays insufficient attention to process and context variables [7]. Furthermore, a systematic review of process variables in research into organizational stress management interventions showed that only half of the included studies had any reference to process evaluation [8]. Without information given by process evaluations, it is difficult to make conclusions about the fidelity to the chosen intervention, the internal validity of the intervention and the generalizability of the results to other contexts [5]. Furthermore, systematic literature reviews need the information from process evaluations as too big variations in included studies regarding what is actually implemented may bias the conclusions of the systematic reviews [9]. 

The study described in this paper is a process evaluation of a randomized control trial (RCT) in the Swedish primary health care system [10]. It will present the results of the process evaluation of implementing an organizational intervention aimed at stress prevention within the primary health care, filling the gap in a mainly effectiveness-oriented research area. It will contribute to the overall understanding of the implementation process of an organizational intervention within the health care setting and exemplify the use of specific frameworks in research into organizational stress management interventions [5].

### 1.1. The Logic Model and the Theory Behind the Intervention Implemented in the RCT Study—The Bases for this Process Evaluation

There is moderate to strong research evidence that high job demand, low job control, low co-worker support, low effort-reward balance and low relational and procedural justice are risk factors for stress-related ill-health [11,12]. According to one of the most researched theories of work-related stress, the demand-control theory [13], reducing demands and/or having more job control will reduce job strain thereby preventing work-related ill-health. Organizational-level participative interventions aim, inter alia, at increasing job control which can have positive effects on employee health [14,15]. Another well-researched theory of work-related stress, the effort-reward imbalance theory [16] describes how the employees’ perceptions of the “cost” and the “gain” in the work context can affect their well-being. High effort and low reward can, according to this theory, lead to negative emotions and to stress response. In the long run, it can have negative effects on mental health [17].

The aim of the RCT was to examine the effectiveness of a participatory workplace intervention as a method to prevent stress. The participatory intervention examined was the Productivity Measurement and Enhancement System (ProMES) [18,19]. ProMES is a participatory organizational-level intervention primarily developed to increase productivity by enhancing motivation [18,19]. However, it is based on research into motivation, feedback, goal setting, participation, role conflicts, and team-efficacy [18], and compatible with both job-demand control and effort-reward imbalance models of work-related stress. 

The main strategies of ProMES address work-related factors known to be risk factors for stress-related ill-health: absence of influence and control; insufficient interaction with co-workers; unclear and conflicting tasks; insufficient participation in decision-making; effort-reward imbalance; insufficient feedback [11,12]. The main hypothesis in the conducted RCT [10] was that ProMES, through participation, defined job demands, set goals, a developed evaluation system, information sharing and increased interaction with co-workers would reduce perceived levels of job demand or increase perceived levels of job control, thereby reducing job strain, in accordance with the theory of work-related stress by Karasek and Theorell [13]. Furthermore, we hypothesized that ProMES increases perceived levels of reward, thereby improving effort–reward balance, in accordance with effort-reward theory of work-related stress [16]. Finally, we hypothesized that ProMES improves sleep and recovery, and in the long term, reduces levels of exhaustion and depression. Table 1 displays the logic model of ProMES as used in the RCT. For a full description of the intervention see the practical guidelines for ProMES [19]. For the more detailed discussion about ProMES and the logic model of the RCT see the effectiveness paper [10].

Our effect evaluation of ProMES [10] found no statistically significant differences between the intervention and the control group for any of the primary and secondary outcome variables (job strain, effort-reward imbalance, exhaustion, sleep, recovery). In other words, the hypothesis that this participative organizational intervention might prevent stress was not supported. Nevertheless, an effect was found regarding reward: even though the workload was higher in the intervention group, employees with no signs of exhaustion at baseline reported at follow up that their work was more rewarding. This was not the case in the control group. In other words, the effects of the intervention differed depending on the level of exhaustion, as measured by the questionnaire at the baseline. Furthermore, even though objective work demands were higher in the intervention group and increased continuously throughout the study, the perception of job strain did not increase. This could imply that the intervention had some preventive effect. The results are described in detail in a separate paper [10]. By conducting a process evaluation of the implementation of the intervention in the RCT, we wanted to contribute to the understanding of whether this lack of effectiveness on stress was the result of an effectiveness failure, or an implementation failure.

### 1.2. Theoretical Frameworks in Process Evaluation

According to Gaglio and Glasgow [20], different evaluation approaches in implementation research have a lot in common. All evaluation approaches emphasize that the implementation of any intervention is a complex process on multiple levels. Furthermore, they emphasize the importance of factors such as context, reach and engagement of the target population. In addition, the assessment of intervention fidelity as well as the assessment of adaptations to the local context are emphasized. Furthermore, Gaglio and Glasgow [20] recommend selecting a framework or combination of frameworks that can give guidance in answering specified research questions. 

For the process evaluation, we followed the UK Medical Research Council (MRC) guidance for process evaluation of complex interventions [5]. Figure 1 displays the key functions of the study’s process evaluation and the frameworks used.

To explore the context, we used the Consolidated Framework for Implementation Research (CFIR) [21], while our description of the implementation process and the mechanisms of impact were guided by the frameworks of Linnan and Steckler [22] and Proctor et al. [23]. 

According to the MRC guidance, the function of a process evaluation differs for the different stages of the intervention research process (i.e., development, evaluation, post-evaluation and scale-up). At the evaluating effectiveness stage, its function is to examine “the internal validity of conclusions about effectiveness by examining the quantity and quality of what is delivered” [5]. As ProMES has not earlier been evaluated for its possible stress-preventive effects, we explore the study participants’ views about the feasibility and acceptability of the intervention, as well as the intervention fidelity and the contextual factors which influence the intervention implementation (facilitators and barriers). 

A meta-analysis of 84 field studies of ProMES [18] identified factors that influenced its effectiveness in terms of its effect on productivity. Some of the factors found to affect the effect size in these productivity studies were: degree of compliance with the ProMES practical guidelines (henceforth referred to as the manual); quality of feedback; how feedback meetings were conducted; and whether design teams had been able to devise an optimal system from the start (i.e., if no changes had to be made in the system after implementation). Contextual factors such as number of people in the workplace unit, turnover of personnel, complexity of ProMES, management support and stability of the organizational environment were not related to the effect size [18]. In the present study, we will assess a broad range of contextual factors that may have influenced the ProMES implementation process. In other words, we want to shed some light on the results and inform future research. The results will indicate what modifications might be necessary for successfully implementing ProMES or other similar group or organizational-level interventions in the future.

The aims of this process evaluation are:(A) To explore the implementation of an organizational intervention, ProMES, as a stress-preventive intervention at a Swedish primary health care (i.e., recruitment, reach, dose). (B) To assess the extent to which the intervention components were implemented in accordance with the original guidelines (fidelity and adaptations);To examine the influence of contextual factors on the implementation process (hindrances and facilitators);To examine the participants’ experience of working with the method (satisfaction, relevance, acceptability, feasibility).

## 2. Materials and Methods 

The population in the process evaluation study consisted of the 67 employees in the intervention unit, employed at the end of the study.

### 2.1. Study Design

We used a mixed method design for the process evaluation [24]. It employed both qualitative (i.e., administrative data, checklists, interviews) and quantitative data, (i.e., a questionnaire). There are several major mixed method designs, such as convergent parallel, exploratory sequential and embedded. [25]. This study used a convergent parallel design as the quantitative and qualitative data collection was concurrent, the components were given equal weight and the two data sets were analyzed and compared. The results of this process evaluation are used for the post-hoc explanation of the findings in the RCT [10]. 

### 2.2. Implementation Object (ProMES)

The core-components (activities) of ProMES are described in Table 1, following the predefined ProMES manual [19]. The manual also defines activities related to the conditions that should be in place prior to starting any ProMES project (activity 0 in Table 1), and a checklist is linked to these conditions [19]. Examples of such conditions are the consistency of ProMES with the organizational reward system; benefits and costs explained to all; management support; organizational resources, such as time granted for working with ProMES; decisions about what software that will be used, etc. 

In ProMES, a distinction is made between the development phase and the implementation phase. The development phase includes activities such as holding initial meetings with the unit personnel, developing objectives, indicators and contingencies as well as receiving management approval—i.e., activities 1–7 in Table 1. In addition to this, time is needed to prepare meetings, solve problems, train supervisors for feedback meetings, etc. The implementation phase of ProMES [19] includes activities such as the continuous gathering of indicator data, regular preparation of feedback reports and regular feedback meetings (steps 8–11).

ProMES is a labor-intensive intervention [19], but there is no predetermined amount of time for its development and there is great flexibility and variation in how it is conducted—i.e., the number of design teams, the number of members in design teams, the number and duration of meetings, etc. Furthermore, as participation is one of the key features of ProMES, the dissemination of information to employees not in the design team is crucial.

### 2.3. Measurement and Data Collection 

Table 2 displays all the process evaluation items, examples of questions and methods of measurement.

#### 2.3.1. Implementation 

*Recruitment, Dose delivered and Reach*: Recruitment is the procedure used to recruit units and employees in the study [22]. Dose delivered tells us how much of an intervention was delivered [22]. Dose delivered was defined as the number of information meetings, workshops and design team meetings delivered by the consultant. This information was collected by administrative data (participation lists). In addition, the duration of the meetings was assessed. Reach is the “extent to which a target audience comes into contact with the intervention” [5]. For the assessment of reach, we used administrative data, logbooks and two questionnaire items: “How would you describe your participation in ProMES?” (response alternatives: 1 = totally passive; 2 = somewhat passive; 3 = neither passive nor active; 4 = active; 5 = very active) and “Have you participated in any design team?” (response alternatives: Yes/No). 

*Fidelity and Adaptations*: Fidelity is the degree to which the intervention components were delivered in accordance with the prescribed manual [23]. Adaptations are the alterations made to an intervention during the implementation and can be purposeful (fitting intervention to a setting) or unintended (i.e., due to barriers) [5]. In this study, we assessed intervention fidelity and adaptations made by using both quantitative and qualitative data—i.e., three checklists, study logbook and information from interviews. Information given during the interviews was used to confirm the information from the study log. 

The checklists were completed by the ProMES consultant and by the unit manager. Checklists were developed by the research team on the basis of the existing ProMES manual. All items are displayed in Appendix A. The checklists were translated into Swedish by the first author and translated back to English by another researcher at the research unit. Checklists I and II assessed whether the activities related to the preconditions and resources described in the manual [19] were in place before starting ProMES. Checklist III was applied after the units had worked with ProMES for six months. It assessed whether the activities and management support, resources, setting, information for employees not engaged in the design teams, etc. remained appropriate during the first six months. The questions also provided information about how far the system had progressed.

#### 2.3.2. Mechanisms of Impact: Participant Experience 

Participant experience and responses to ProMES were assessed by a questionnaire administered to the 67 employees one month after the RCT 12-month follow up. The questionnaire contained in total 22 questions based on the taxonomy by Proctor et al. [23] and Linnan and Stecklar [22]. Questions assessed the appropriateness/usefulness, acceptability, satisfaction (with the content, work procedures and the delivery of the method), sustainability and feasibility of ProMES. Furthermore, the questionnaire questions were based on the ProMES questionnaire used in the ProMES meta-analysis [18]. We used and adapted questions primarily from the “Reactions to the system” part of the questionnaire (things that participants liked and disliked about the system). Appendix A displays all the questions in the evaluation questionnaire as well as all answer alternatives. Appendix A displays how the questions in the questionnaire relate to the taxonomies of Proctor et al. [23] and Linnan and Steckler [22].

#### 2.3.3. Context 

Contextual factors may affect the implementation, the intervention mechanisms and the outcomes [5]. In the present study, we conducted one focus group and three individual interviews to assess the context. All interviews were completed two months after the 12-month follow up. The two-hour focus group was moderated by the first author (an experienced licensed psychologist), with the third author as a co-moderator. Individual interviews were conducted by the first author alone and lasted one hour. We used purposeful sampling to select the informants, as qualitative research emphasizes deeper understanding and information saturation rather than generalizability and representativeness [26].

For the focus group interview, we chose a maximum variation strategy—i.e., the focus group participants were representatives of the seven design teams, one for each occupational group. This enabled us to collect information on how different occupational groups experienced ProMES. For the individual interviews, we used the extreme case/intensity strategy, which also emphasizes the variation but illuminates cases at the “extremes” and their diverse experiences [26]. In this study, we selected (1) a representative of the occupational group that had the longest progression in their work with the intervention (i.e., started with the feedback meetings), (2) a representative of the occupational group that had not come so far and (3) the manager as a representative of the organization’s leadership. In the results section, quotes are anonymous for ethical reasons.

### 2.4. Data Analysis

Quantitative data—i.e., questionnaire data about participants’ experience of ProMES—were first analyzed using descriptive statistics. As participation is one of the cornerstones of ProMES, and our RCT studied a participatory organizational-level intervention, in the process analysis, we also wanted to explore whether there were any differences between the active and less active employees (i.e., if participation affected the experience of working with ProMES). If an employee answered “yes” to the question “Have you participated in any of the design team’s work?” and chose “active” (4) or “very active” (5) as a response to the question “How would you like to describe your participation in the work with ProMES”, the employee was defined as “active”. The rest of the employees were defined as “less active”. The Mann Whitney test was used to compare active and less active employees’ experience of ProMES. We used IBM-SPSS Statistics, version 25 (Karolinska Institutet, Stockholm, Sweden), for the statistical analysis.

We used content analysis for the qualitative data, [27,28]. Because we worked deductively [29], the domains and categories were taken from the theoretical model—in this case, Consolidated Framework for Implementation Research (CFIR) [21]—which assesses the context by examining the barriers to and facilitators of implementing intervention. It is composed of five domains: intervention characteristics, outer setting, inner setting, characteristics of the individuals involved and the implementation process. Each domain comprises several constructs.

The analysis comprised the following steps. Each interview was read through several times to get a sense of the content. All the text that concerned the purpose of the analysis (facilitators and hindrances) was highlighted and copied into a separate template (step 1). The meaning of each text paragraph was accordingly condensed (step 2). The condensation was as close to the original text as possible. During Step 3, a “Code sheet” was created. The condensed text from Step 2 was copied into the code sheet. We assigned each text paragraph (i.e., its condensed meaning) to the appropriate domain from the CFIR. Subsequently, the appropriate constructs (sub-domains) from CFIR were assigned. We used the same procedure for each interview. 

In Step 4 we used the “CFIR memo template” [30]. The condensed text was placed in the memo template under each domain/subdomain in accordance with Step 3. Since page references are always given, it is possible to trace back the condensed text to the original interviews. The key words in the memo template were then highlighted. From here the material was further condensed to find the various “themes” that could appear in the subdomains, again as close to the text as possible. For example, in the intervention characteristics domain, there might be a text in the subdomain relative advantage, which indicates that the unit has achieved better cooperation or a better structure as a result of working with the intervention. This would then be regarded as two different themes in this subdomain (cooperation/structure). 

Two researchers then independently rated the valence of each coding—in other words, whether the influence on implementation was positive (facilitating) or negative (hindering)—according to the CFIR rating rules [31]. In the last step, the results were summarized and described. In deductive analysis, the “rest” (i.e., what is left after the above described process) goes to the inductive part of the analysis. If the material gives rise to new categories not originally found in the selected model, they are also described. No new categories were found in this material.

## 3. Results

### 3.1. Implementation Outcomes

#### 3.1.1. Recruitment

The research team contacted a large primary health care division of a Swedish county council. The division consisted of 29 primary health care units located at different geographical locations in the same county council municipality. All the units were informed about the study by their management and occupational health service. Four units participated in the study. The other units were not eligible for this study as they were already participating in other activities at division level. All employees in the units were involved in the decision to participate. 

The three units were then randomized into one intervention and two control units. The intervention unit consisted of 57 individuals at the start of the study and 67 at the end of the study. Each professional group (nurses, physicians, physiotherapists, etc.) formed their own design team comprising two to four representatives. In total, seven occupational design teams were formed. Members of professional groups were free to rotate as representatives of their group in the design team.

#### 3.1.2. Reach

Of the 67 employees, 49 (73%) answered the process evaluation questionnaire. Of the 18 non-responders, one was on parental leave, four had started a professional training one month prior to the end of the study, four were newly employed (1–2 months before the study ended) and three were employed by the hour. For the remaining six employees, we have no information about their reason for not answering the questionnaire. Of the 48 employees who responded to the question about active participation, 52% considered themselves as being actively or very actively involved in ProMES. Another 40% indicated that their participation was neither active nor passive. Four percent described themselves as “somewhat passive”, and only 2% saw themselves as completely passive. Thirty of 49 respondents (61%) who responded to the question about working in the design teams answered that they had participated in one of the design teams at some point. Twenty employees (42%) described themselves as both active and very active and had participated in the work of the design teams.

#### 3.1.3. Dose Delivered 

The consultant held an hour-long information meeting for all employees in May 2013, but the study started formally with a full-day workshop with all employees in September 2013. During the workshop, ProMES was explained in more detail, the mission and the vision of the unit were discussed and the work with the overall objectives started. The purpose of the workshop was both specific (i.e., to work on the objectives) and general (i.e., to facilitate the implementation of the intervention by increasing participation). 

Seven design teams were then formed—one for each occupational group. Design teams continued to work with the consultant during October–December 2013. At the units’ regular occupational group meetings and workplace meetings, the design team members discussed their work with their colleagues. The second full-day workshop with all the employees was held in December 2013. During the workshop, the occupational groups reported on their work and progress, and continued working on objectives, indicators and contingencies. The work in design teams and information sharing at workplace meetings continued January–September 2014.

The consultant held in total 33 meetings with the seven design teams. The number of meetings, duration and starting point varied per design team (one to seven meetings). The meetings lasted from half an hour to (most often) one to two hours.

The participation rate for the two workshops was 93% (53/57). The participation rate for the workplace meetings varied between 60% and 74%. In addition, the manager and the consultant had continuous planning and preparation meetings (according to the study logbook, at least four such meetings took place between May and August 2013). The consultant spent a total of 528 h (during the year) working with the unit. 

#### 3.1.4. Fidelity and Adaptation 

We assessed fidelity in terms of fulfilling the core components (activities) of ProMES according to the steps described in the manual. 

The main results showed low fidelity regarding the implementation of regular feedback meetings (steps 8–11 in the logic model). According to the manual, feedback meetings should have been held at 6 months after the start of the intervention. No occupational groups had, however, participated in regular feedback meetings with written feedback reports on all indicators at six months. There was a variation between the occupational groups. Some groups received feedback on some indicators (for example: number of dictations—medical secretaries), while others did not receive any regular feedback. An indicator that was measured regularly and fed back to all the employees was availability (answering patient calls). Patient satisfaction was an example of an indicator that the groups sometimes received feedback about.

Regarding adaptations made during the study, one important adaptation was the formation of not one but seven design teams, one for each occupational group. This adaptation was made by the facilitator in an effort to increase participation. When organizational units are larger than fifty employees, it is not unusual to have separate ProMES systems for each subgroup—i.e., separate objectives and quantified indicators [18]. It is unclear how this effected the implementation process.

The checklists, completed by the unit manager and the consultant, showed high fidelity to the manual regarding the general preconditions before starting ProMES—i.e., agreement that 11 of the 14 activities described in Checklist I—were performed. There was a disagreement whether support existed on all management levels, whether the issues of compensation if productivity goes up were dealt with and whether productivity measurement should be linked to pay if production goes up. They also agreed that the resources needed for the development and the implementation of ProMES were in place before start (agreement on nine of 12 activities in Checklist II). In Checklist II, there was a disagreement regarding whether enough resources were provided for training in the use of software (web-based IT-support for working with ProMES), whether a decision had been taken about when and how the organization would take over the preparation and whether the distribution of the feedback reports and facilitator time outside the meetings were permitted. 

Checklist III shows that the activities related to a formal process to give information to the members of the target unit who were not part of the design team (memos, announcements, rotating personnel through the design team, etc.) were performed, although not on a regular basis. Other activities related to the conditions for the successful implementation of ProMES were fulfilled. Examples of such conditions are whether consensus was reached in meetings on most major issues and whether the system was explained to the entire unit at a meeting (workshop).

### 3.2. Mechanisms of Impact: Participants Experience 

The response rate to the process questionnaire was 73%. In total, 90% of respondents agreed or strongly agreed that ProMES clarifies what is important and provides an opportunity for improvement. Furthermore, 78% agreed that ProMES increases employee participation in decision-making, and 74% agreed that it gives employees more control. They were satisfied with the management support (76%) and the consultant support (82%), and 84% agreed that ProMES increases work efficiency. Fifty-six percent wanted to continue working with ProMES, and 66% reported that ProMES is good for reducing work-related stress. Fifty-seven percent reported that it is easy to maintain, while 72% considered it to be time consuming. Appendix A displays all the results of the process evaluation questionnaire. 

There were no differences between how active employees and less active employees responded to the questionnaire. The only exception was the question on satisfaction with ProMES. Active employees were more satisfied than their colleagues that were less active (Z = −1.988; *p* = 0.047).

### 3.3. Context Facilitators and Hindrancies 

The results of the interviews showed that the majority of the facilitators and hindrances influencing the implementation of ProMES can be categorized into the CFIR domains intervention characteristics, inner setting and process. This section is structured according to these domains. To increase readability, we have made some minor adjustments to the quotes below, but these changes do not affect their meaning.

#### 3.3.1. Intervention Characteristics 

The interviewees were overall positive about ProMES. They indicated that ProMES gave more structure and that ProMES helped them to reflect on “the whole picture” and not only on their own occupational group. The interviewees reflected on how their work affected their colleagues:

“I have been exposed to many such methods over the years, but this is the first one that looks at the big picture… Before (prior to ProMES), the doctors would come up with things that gave more job for nurses or for laboratory technicians or secretaries, and especially the secretaries. This is the first time we have looked at the big picture”.

ProMES also facilitated collaboration within the occupational groups as well as the distribution of tasks. It was mentioned that the measurements/statistics resulting from ProMES provided an overview, for example of their workload, and an opportunity for better planning and documentation. The results gave an opportunity to adjust working procedures. Interviewees felt that ProMES improved the feedback process and helped to create a good working atmosphere. However, some interviewees were uncertain about whether the effects they saw were a result of working with ProMES or of other changes that had started before ProMES:

“We may have answered more phone calls, increased the number of visits…. But if it is because we have implemented ProMES, or if it is because we have strengthened the staffing or just a natural result of our own usual development work, I cannot say one or the other”.

Some interviewees were critical towards the measuring involved in ProMES. They thought, for example, that measuring the number of phone calls or patient visits was irrelevant as visits could be short or long, and patients easy or complex. They wanted quality to be measured instead. “Before we started measuring all kinds of things, we should have reflected on why we measure things, what do we want to get out of it? I get the feeling that we only measure for the sake of measuring. Where maybe we should have been more careful and thought about it more: What is the value of measuring this?” Furthermore, ProMES was seen by some as time-consuming, by others not at all. Some understood that working with ProMES was not meant to be a time-limited process but a sustainable part of their working process, while others requested an end date. Furthermore, some serious concerns about dependency on the consultant and his data program were expressed, because the organization did not have access to the consultant’s data program during the study. Some people had difficulty understanding ProMES and interpreting the graphical presentations that were produced via the web-based support:

“If the consultant is not here to help us anymore…I think of these tables when I pass the board, I think there is a little too much to read, one cannot take everything in. I think that if we are in it without the consultant then you want it collected in a simpler way so that we can quickly see what we have done”.

#### 3.3.2. Inner Setting

Available resources and structural characteristics were the two CFIR subdomains within the inner setting domain that appeared frequently as hindrances. The most prominent themes within these subdomains were shortage of staff and the resulting time pressure; staff turnover; difficulty in obtaining the necessary administrative data from the central administration office; and overcrowding. Difficulty in obtaining the data from the central administrative office was mentioned most frequently: 

“We have met the primary health care management at higher level and the person responsible for IT in primary health care, and the data output group at county council level, and they do not think that the outputs we are interested in are interesting enough to invest in…I think it is wrong to think that we can only go on measuring what the politicians or the leaders of the county council want to measure, wrong not to invest in the fact that the working people have thoughts and ideas”.

The work with ProMES was also severely hampered because chosen indicators could not be measured due to problems with obtaining the data from the administration office. Continuous feedback meetings could not be held regularly in accordance with the ProMES guidelines. Furthermore, subgroups and especially design team leaders had to do extra work collecting data manually (as employees had to do their own weekly statistics). Another hindering factor was shortage of staff in some subgroups, in part due to planned retirements:

“Because when we get below the 50 percent (of usual workforce), then it starts to be… Not good. Then, it does not matter which system you use. There are just too few hands”.

The consequences were that, periodically, there was no time for ProMES because of patient overload. Poor staffing and time shortage were also given as the main reasons for team leaders having difficulty collecting statistical data from their colleagues. Some occupational groups were not affected by understaffing, and these subgroups came far in their ProMES work. Furthermore, high staff turnover meant it took some time for new employees to get to know each other, restructure the group’s work, supervise the new employees and introduce them to ProMES. This took time from regular duties as well as from the ongoing work with ProMES:

“Such a large staff loss was not even in our wildest imagination. In addition, new nurses and physicians under specialist training required extra supervision”.

“Personnel turnover is a disadvantage because those who have participated in the process disappear, and those who come are not on the train, they do not know what it is about”.

Another hindrance was the increase in patient numbers. The number of patients on the intervention unit’s list increased steadily, both before and during the study (for example, 4.2% from the baseline to the 6-month follow up). The number of employees also increased during the 12-month period. As a result of the increase in patient and employee numbers, premises became too small, which was a further problem. 

An important facilitating factor in the inner setting domain was an engaged first line manager, who also communicated clearly about the importance of everyone´s engagement.

#### 3.3.3. Process

The important role of the consultant was emphasized several times. He was described as accessible and committed, motivating and enthusiastic. He gave practical help, commented on their work and initiated work on some difficult issues.

There was some variation between subgroups with regard to how engaged key stakeholders (unit members) were. This was due in part to group size. In some groups, everyone was engaged, and they discussed and decided everything together. This was more difficult in bigger groups:

“I can feel that we are a fairly large group, there are 15 of us and there were 4 who were working with the consultant. I think we others did not have the same participation as those who sat with him, that there was not so much time to address everything that was expressed in the group. Then the consultant sent out what had been said (in the design team) and how it improved and so, so it is good to have taken part … but not everyone in our group was so involved…”.

Some felt that they had learned a sustainable working model that helped them to structure their work and solve problems. Furthermore, noticing progress boosted their willingness to continue. After a while, however, when they noticed that the results were similar from week to week, some felt they had lost the impetus to give regular feedback. Overall, working with ProMES in the separate occupational subgroups was perceived positively. However, it was felt that there were too few meetings between the subgroups and not enough time for meetings with the entire unit. Interviewees said that they knew too little about what the other subgroups were working with and how their own work affected the other groups.

“It is still the case that it is not the solution to the staff shortages, but I think that with the model (i.e., ProMES) and especially if we were to have more meetings between different professional categories, then I think that we would gain more for the health care unit, than just working with each individual professional group”.

There was quite a lot of discussion about the future. During the study, the unit was also involved in preparations for future reorganization into three care teams. Some thought that the reorganization could contribute to more spin in work with ProMES, as future teams would have fewer employees. Others were keen to compare future teams:

“When we have three care teams, we will also have a greater opportunity to compare different teams with each other based on different working methods. To test different stuff. We have all learned a model that we can build on. We hope that the work will continue”.

## 4. Discussion

The aims of this process evaluation were to explore the implementation process of an organizational intervention, ProMES, in a primary health care unit in Sweden, to examine the influence of contextual factors on the implementation process and to examine participants’ experiences of working with ProMES. The study showed that even though the reach was satisfactory, more than half of the employees were actively or very actively involved in working with ProMES and the employees had a positive attitude towards ProMES, it was not fully implemented as intended. The unit never reached the point of holding regular feedback meetings as described in the manual. On the basis of the contextual factors identified in the study, we will now discuss which factors might explain this implementation failure before discussing some theoretical implications.

One barrier that may have influenced the implementation of ProMES is related to the CFIR’s inner setting domain. The results showed that there were difficulties in obtaining the statistical productivity data from the central administration office, and these data were vital for the development of contingencies (i.e., the operationalization of the relationships between the results and the evaluations). Due to the lack of statistical data, design teams were unable to develop full feedback reports and have continuous feedback meetings. The difficulties in obtaining statistical data might be attributable to the reported lack of support from the highest management level, which likely meant no support from the central administrative and IT levels. Several previous studies have identified management support as an important facilitator of the successful implementation of intervention studies [32]. 

Other barriers associated with the inner setting that may have influenced the implementation process were related to the availability of resources. For example, there was a shortage of staff in several occupational subgroups, and this resulted in time pressure. The high turnover rate added an extra burden in terms of providing support and introduction to new employees, which likely resulted in even more time-pressure and less time for working with ProMES. Moreover, the intervention unit had a steadily increasing inflow of patients and a shortage of physical space. Staff turnover and high workload are known to be barriers to the implementation of any intervention [33,34]. The staffing in the intervention group was not seen as a problem at the start of the study—i.e., that precondition was met. However, problems accelerated during the study. It is likely that the shortage of staff resulted in more job-strain for the employees and thereby counteracted the intervention because less time could be devoted to ProMES. Similar findings regarding personnel shortages as a factor which hinders the use of a stress preventive strategy were found in a study of a digital platform-based implementation strategy aimed at stress prevention [35]. Adequate staffing is a basic resource that needs to be in place during the whole intervention period and is crucial to the successful implementation of any intervention, or for that matter any change management. 

Adding resources such as participation in decision-taking, better feedback, social support, control over goals or evaluation system, control over breaks, etc. will not buffer unreasonably low staffing. In a systematic review of how resources improve employee health and performance [36], researchers conclude that resources at any level (individual, group, leadership, organization) are related to employee wellbeing and performance. The resources examined at the organizational level were job characteristics (such as autonomy), HR practices (such as training, performance appraisals, etc.), fit between person and organization, etc. Staffing, which reasonably can be seen as a basic organizational resource, seems not to be included. Furthermore, the shortage of staff can be seen as an involuntary downsizing, and the effects of downsizing on health are well known [37]. The autonomy and other resources, as described in the review of Nielsen et al. [35], may be important when basic needs are already satisfied. It is possible that with full staffing, the results of our RCT [10] would be different. 

The importance of having the right prerequisites (such as enough staff) for implementing ProMES was clearly reflected when occupational groups were compared with each other. Those occupational groups which stated that they were fully staffed (and, therefore, had more time to work with ProMES) progressed further, participated more and had fewer problems collecting the individual data on their own. Earlier research supports this interpretation and demonstrates that work resources are important for increased engagement in the implementation of organizational changes [38]. Unfortunately, the number of employees in the occupational groups was too small for any statistical comparisons. For ethical reasons, it was also too small for more detailed descriptions. However, given the results in our RCT study [10] regarding the differential impact of the intervention (depending on the scoring on the exhaustion scale at the baseline measurement) and the above reasoning, it is an important ethical question when and how to implement an intervention, including organizational change interventions. Individual support during the organization change may be needed, in particular if resources such as staffing are restricted.

Other factors which may have hindered the implementation process, were related to intervention characteristics and include the two CFIR subdomains complexity and design and quality of packaging. Where complexity was concerned, ProMES was seen as time consuming and new employees found it difficult to understand. With regard to design and quality of packaging, employees had difficulty interpreting the graphical presentations and understanding all the available charts. The complexity and design issues of ProMES may explain the reported low satisfaction. According to the survey, only half the respondents said they would like to continue working with ProMES. Before starting ProMES, it is, therefore, clearly necessary to establish that it is possible to access and collect existing organizational data. It is also necessary to clarify the amount of training needed to understand the software. Moreover, seven design teams were formed in this study. This enhanced participation but also had some negative consequences: as there was only one consultant, the work could neither proceed simultaneously in all the occupational groups nor be completed before the study period ended. 

Few of the respondents strongly agreed that ProMES is a good method for reducing their work-related stress. These results are not surprising given that the staffing shortages and staff turnover described above are contextual factors that cannot be solved by ProMES, but rather by either recruiting more personnel or reducing the workload. This is in line with research showing how important it is to focus on job demands such as work pace and work load, because merely improving resources is not necessarily enough as a buffer against future burnout and sick-leave [39]. In other words, adding resources such as participation in decision-making, better feedback, control over goals or evaluation systems will not counteract the effects of unreasonably low staffing. Reasonably, staffing can be seen as a basic organizational resource, and a shortage of staff (for whatever reasons) can be seen as an involuntary downsizing. The negative effects of downsizing on health are well known [37,40]. 

To enhance both the implementation and the effectiveness of ProMES or any other stress-preventive intervention, tighter integration might be needed with other organizational systems (i.e., administrative, employment practices, managerial system)—a finding which is in line with a systematic review of intervention studies aimed at improving well-being and performance [41]. Neoliberal ideology prevails in management practice [42], and new public management is an integral part of the health care system in Sweden. Neoliberalism and new public management lead inter alia to “a focus on quantitative assessment, control and monitoring…” [42]. If integrated with other organizational processes, ProMES can give employees some control over assessment and monitoring, thereby possibly mitigating some of the negative effects of new public management and achieving less stress and more “dignity” [42]. However, the Swedish health care system has, during the past decade, implemented lean thinking with the aim to improve quality of care and increase efficiency [43]. In other words, a lot of measuring besides that initiated through working with ProMES was taking place during the study and it is unclear how these two processes interacted with each other.

Furthermore, as some interviewees in our study expressed, qualitative aspects of work may get lost in all quantitative measurement. In accordance with some research [44], strong professional identity may be threatened by illegitimate tasks which are then perceived as stressors. Illegitimate tasks are those tasks that are seen by employees as unnecessary or unreasonable. In Sweden, some of the objectives of the health care system are politically controlled [45] and beyond the control of supervisors. For example, due to politically placed requirements for increased availability, giving time to patients with simple ailments could, by a professional, be seen as less important than taking time with a chronically ill patient with a complex disease. It may lead to perception of availability as an illegitimate task, which can turn into frustration, particularly if staffing is a problem. Quantitative measure of availability could in that case be seen as “illegitimate” even if the group participated in the design of effectiveness curve as a part of ProMES work. It can also be expressed as a “friction between the goals set by the organization and someone´s self-interest…” [46]—in this case, the care-professional´s identity.

In addition, organizational interventions such as the one described in this study are almost exclusively focused on job resources (control, participation, feedback, support, etc.). Interventions that are focused on decreasing objective job demands by “having more people do the same tasks, giving more time per person to do the same tasks or reducing the number of tasks per person” [3] are rare, despite the knowledge that the high workload is among the major problems in health care [47]. One reason for this is probably to be found in the political and societal changes of our time such as declining resources and new public management [48]. Another reason may be that there is a limited knowledge on how to manage workload through work system redesign in a such extremely complex and changing system [47]. Carayon et al. (2011) are, therefore, calling for more collaborations between sociotechnical system analysts and clinicians to understand workload in different settings. 

Furthermore, in a protocol for a Cochrane review regarding organizational level interventions for reducing occupational stress in healthcare workers [3], researchers are categorizing organizational interventions into six groups—i.e., interventions that (a) decrease job demands, (b) increase job control, (c) improve workplace social support, (d) improve clarity in work tasks/roles, (e) enhance work processes and (f) improve organizational communication. According to theory, ProMES seems to fit into all of the above. In other words, ProMES (a) gives a certain degree of participative decision making with regard to demands—i.e., which work results need to be improved and how much; (b) increases control through participation in the design of the feedback system and “provides feedback with regard to what employees need to start doing, stop doing, or continue doing…”; (c) improves social support through group reflexivity and cooperation; (d) reduces role ambiguity by identifying the results-to-evaluations connections; (e) enhance work processes through improving team effectiveness; and (f) enhances cooperation and coordination through meetings (horizontal communication) and through approval of goals, indicators and contingencies through approval from the management. (vertical communication). Participants in our study seem to agree with this, according to the process evaluation questionnaire described above. Despite the capability of ProMES to address all organizational intervention points, we could not discover clear effects on stress experience. One explanation for that could be that the workplace resources addressed by ProMES are on only one level. According to newer research [36], interventions targeting multiple resource levels (individual, group, leadership and organizational, IGLO) are due to their synergistic effects to be preferred when targeting employee well-being. However, to decide which combination of interventions is most appropriate in a certain context, a thorough analysis of needs should precede any intervention. 

Moreover, a study of stress prevention needs of employees and supervisors [49] shows that both employees and supervisors indicate the need for more attention for job demands, and that job demands often depended on factors beyond supervisors control such as personnel shortage (or politically made decisions as described above). In other words, another explanation for the small effects on stress could be theory failure—i.e., improving only resources may not be enough. According to the new meta-analytic review of longitudinal studies of the job-demands resources model [50], which is now by far the most used framework in the work health research area, there is a need to address a differentiation between hindering and challenging job demands as well as between social and structural job resources. Inappropriate staffing can be among the causes for perception of job demands as hindering and appropriate stuffing can be regarded as a structural resource. In the light of Bal and Doci (2018) reasoning about neo liberal ideology and work and organizational psychology (as well as occupational health psychology) and in line with our results, it is interesting to notice that there is so little attention on hindering, objective job demands (including workload) and much more emphasis on job crafting, personal resources and job performance than on the responsibilities of the organizations, higher managements and politicians to design a sustainable work environment with reasonable preconditions for achieving imposed goals.

### 4.1. Methodological Considerations

A systematic review of process variables used in the evaluation of organizational stress management interventions [8] found that there is a lack of use of standardized and comprehensive frameworks in this line of research. A strength of this study was the use of MRC guidance for process evaluation, as it reduced the arbitrariness of our choices, guided our research questions and facilitated the process evaluation. Another strength is the use of the CFIR as a framework for the analysis. The arbitrariness in the analysis was reduced to some extent because CFIR gives a consistent set of constructs and their definitions and, therefore, a more systematic coding. However, more work is needed on “sometimes indistinct boundaries between constructs” [21] to avoid the need for double coding. The third strength of the study is the use of both qualitative and quantitative data in such a way that they validate each other. 

A limitation is that our integration of the qualitative and quantitative research is somewhat incomplete or “partial” [51], because the low number of participants did not give us the opportunity to carry out subgroup analysis based on the findings for the qualitative data. One such analysis would be a secondary analysis—i.e., a comparison of RCT outcome data for the occupational subgroups which had come quite far in their work with ProMES and had held some feedback meetings and those subgroups which had not come so far. Another limitation is that we used CFIR for data analysis only and did not integrate it throughout the process (design, data collection, analysis) as recommended in the systematic analysis of the use of CFIR [52].

### 4.2. Implications for Future Research

A study with a larger sample could shed more light on the effects of the intervention. In addition, there is a need to explore how to protect and support already vulnerable individuals during the implementation of any organizational intervention or change. One way to facilitate the above (as well as to facilitate the implementation of interventions in general) is to invite organizations much earlier in the research process, and establish practice based research networks, as recently described in a Swedish report on a practice based research network approach [53].

## 5. Conclusions

This study demonstrates that ProMES enhances productivity, clarifies priorities, gives employees more control, gives good performance feedback and increases participation in decision-making. However, it is also a work-intensive intervention, and its stress-preventive potential is still unclear. Not unusual with the complex organizational interventions, full implementation was hindered by many factors. The introduction of ProMES higher up in the organizational hierarchy and better integration with other organizational systems would probably have removed at least some of these obstacles. To be successful in stress prevention, organizations need to target hindering job demands in parallel with implementing interventions that increase job resources. Furthermore, when considering implementing an intervention or organizational change, it is important to understand how the amount of change interacts with lack of resources such as staffing, and how it may affect some subgroups of employees by taxing their resources. This study also demonstrated the usefulness of MRC guidance in process evaluation.

## Figures and Tables

**Figure 1 ijerph-17-07285-f001:**
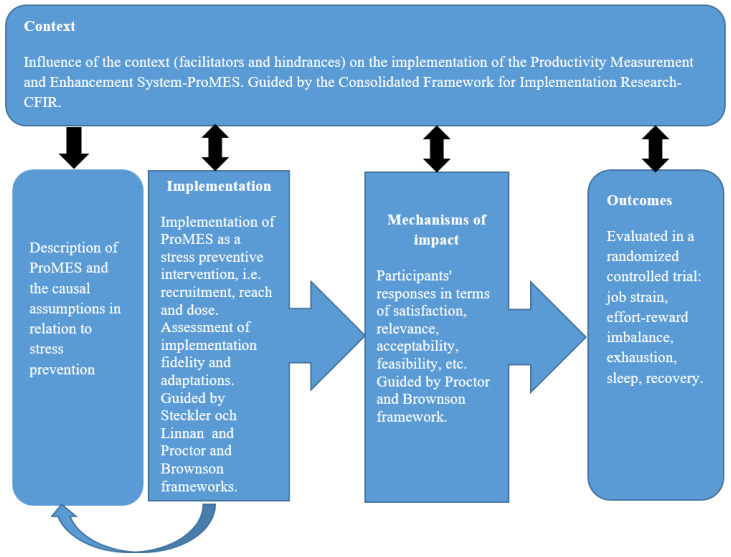
Key functions of the process evaluation and the frameworks used (after UK Medical Research Council (MRC) guidance [5]).

**Table 1 ijerph-17-07285-t001:** Logic model of ProMES/Theory of Change.

Problem and Evidence Base	Resources	Activities/Strategies/Core Components of ProMES	Method	Short-Term Outcomes	Medium-Term Outcomes	Long-Term Outcomes
Preconditions	Management	0. Activities before starting ProMES	Checklists			
Some work-related factors are known to be risk factors for ill-health: absence of influence and control, insufficient interaction with co-workers, unclear and conflicting tasks, insufficient participation in decision-making, low esteem reward, and insufficient feedback	Experienced ProMES fascilitator Multi-disciplinary research team Funding	1. Initial meetings 2. Form design team 3. Identify objectives = identify results (Input from the whole unit. Decision making process is discussion to consensus) 4. Develop indicators=operationalize results 5. Approval from management 6. Develop contingencies = operationalize relation between result and evaluation 7. Approval by management 8. Gather indicator data 9. Develop feedback reports 10. Conduct feedback meetings = information on results and evaluations of results for the determined period of time and if needed improvement/development of new strategies 11. Monitor project over time	(1) Participative decision making with regard to demands, i.e. which work results need to be improved and how much; (2) involvement in the development of the evaluation system, thereby increasing the likelihood of perceiving it as accurate and fair; (3) timely feedback about the results of one’s efforts that can be thought of as a positive reinforcement; (4) information sharing; and (5) discussion of problem solving and work strategies	Controllable and acceptable demands. Higher control through higher decision authority. Higher reward through extrinsic recognition (from pears and management). Better social support (coworkers and supervisors). Other possible mechanisms: Justice (measurement system perceived as fair). Self-efficacy: Confidence in the group’s ability to take action and overcome barriers.	Primary outcome: Lower levels of job strain. Secondary outcomes: Lower effort-reward imbalance. Better sleep and recovery.	Lower occurrence of common mental disorders such as: Depression and exhaustion (secondary outcomes).

**Table 2 ijerph-17-07285-t002:** Process evaluation items, examples of questions and methods of measurement.

Process Evaluation Items	Examples of Specific Questions	Methods of Measurement
Implementation		
Recruitment	Who recruited the participating units and how? What were the reasons for participation and declining participation? What characterizes units that declined to participate?	Administrative data, project logbook.
Dose delivered	Frequency and duration of meetings.	Administrative data, poject logbook.
Reach	What percent of employees in the participating unit participated in ProMES? How would you describe your participation with the ProMES? Have you participated in ny design team? Yes/No	Administrative data, project logbook, 2 questions from questionnaire.
Fidelity and adaptation	To what extent was ProMES implemented as intended (according to manual)?	Project logbook, checklists, information from interviews.
Mechanism of impact		
Participant responses	How satisfied are the employees with the content, work procedures, delivery. Their opinion of ProMES’s appropriateness/usefulness, acceptability, feasibility, sustainability.	Questionnaire.
Context	Which factors/circumstances have either facilitated or hindered working with ProMES?	Semi-structured interviews.

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
