# Peer review of "Process Evaluation of a Participative Organizational Intervention as a Stress Preventive Intervention for Employees in Swedish Primary Health Care"

_ijerph, 2020, doi:10.3390/ijerph17197285_

Round 1

Reviewer 1 Report

As per the sentence in the first paragraph beginning "However", I would recommend that the authors check the following two articles on organizational-level interventions.

Giga, S. I., Fletcher, I. J., Sgourakis, G., Mulvaney, C. A., & Vrkljan, B. H. (2018). Organisational level interventions for reducing occupational stress in healthcare workers. The Cochrane Database of Systematic Reviews2018(4), CD013014. https://doi.org/10.1002/14651858.CD013014

Pignata, S., Boyd, C. M., Winefield, A. H., & Provis, C. (2017). Interventions: Employees' Perceptions of What Reduces Stress. BioMed research international2017, 3919080. https://doi.org/10.1155/2017/3919080

Author Response

Point 1:

As per the sentence in the first paragraph beginning "However", I would recommend that the authors check the following two articles on organizational-level interventions.

Giga, S. I., Fletcher, I. J., Sgourakis, G., Mulvaney, C. A., & Vrkljan, B. H. (2018). Organisational level interventions for reducing occupational stress in healthcare workers. The Cochrane Database of Systematic Reviews2018(4), CD013014. https://doi.org/10.1002/14651858.CD013014

Pignata, S., Boyd, C. M., Winefield, A. H., & Provis, C. (2017). Interventions: Employees' Perceptions of What Reduces Stress. BioMed research international2017, 3919080. https://doi.org/10.1155/2017/3919080

Answer 1:

Thank you very much for your time and engagement in reviewing our paper. We have now added some information and incorporated one of the studies you suggested. Page 1, lines 40-42.

Reviewer 2 Report

Dear authors,

Thank you for your interesting submission! I like your topic very much! Please find my comments below to improve your manuscript.

1. Please try your best to find out one theory to explain why the organizational interventions could effectively address the stress in this study, and further literature review regarding this theory is necessary. For me, how to ensure the interventions are effective from the perspectives of theory or framework is extremely important.

2. Have you ever closed any research gap based on your literature review on the previous studies and relative studies of the theory you wanted to apply in your own study? Clarify your contributions to this research field.

3. Therefore, my suggestion is to re-structure the introduction section and to make it well-structured.

4. Please clarify the sample you wanted to focus on. You have to focus on one particular sample due to specific reasons. Further revision of your research question is also necessary.

5. How did you ensure the quality of your instruments in this study?

6. How did you control the confounders and bias in your statistical analysis?

7. Try to make your presentation of your findings well-structured.

Looking forward to your next version! Good luck!

Author Response

Reply to Comments and Suggestions for Authors

Dear authors,

Thank you for your interesting submission! I like your topic very much! Please find my comments below to improve your manuscript.

Thank you very much for your review! We have improved our manuscript in accordance with your suggestions.

Point 1: Please try your best to find out one theory to explain why the organizational interventions could effectively address the stress in this study, and further literature review regarding this theory is necessary. For me, how to ensure the interventions are effective from the perspectives of theory or framework is extremely important.

Answer 1:

Thank you for your comment. In the effect paper (masked reference) we have elaborated on this in some length. However, we have now tried to make the theories behind the logic of use of ProMES (job demand-control theory and effort-reward imbalance theory) more explicit in the introduction section in this paper too. See Page 2, Lines 75-94, page 3, lines 95-97 and page 5, lines 132-140.

Point 2: Have you ever closed any research gap based on your literature review on the previous studies and relative studies of the theory you wanted to apply in your own study? Clarify your contributions to this research field.

Answer 2:

There is a scarcity of process evaluation in research into organizational stress management interventions which has by several previous studies been identified as a research gap that needs to be filled, see page 2, lines 53-57. We have now added some clarification, please see page 2, lines 58-63. The process evaluation of ProMES contributes to the research field by using the implementation of ProMES as an example of what influences the implementation of a stress prevention intervention within the primary health care. To our knowledge there are no other process evaluations of the implementation of ProMES as a stress preventive intervention.

Point 3: Therefore, my suggestion is to re-structure the introduction section and to make it well-structured.

Answer 3: In accordance with the suggestions we have (in addition to the above) to some extent re-structured the introduction by adding information on page 2, lines 58-73. We have also added another subtitle, see lines 75-76 (which can make the structure of the introduction clearer), changed the order of information in that section, and added lines 96-97.

Point 4: Please clarify the sample you wanted to focus on. You have to focus on one particular sample due to specific reasons.

Answer 4:

Thank you for your point. As this is a process evaluation, the sample consists of employees in the interventions group which we described under subsection reach (results section). However, as you point out, it might not be evident for all readers and therefore, in line with your recommendation, we now added a sentence on page 6, line 178-179.

Further revision of your research question is also necessary.

We find it difficult to change the research question now, after the data has been collected and the analysis has been done.

Point 5: How did you ensure the quality of your instruments in this study?

Answer 5:

As far as possible existing instruments were used. The process evaluation questionnaire was based on the ProMES meta-analysis questionnaire See page 8, lines 243-245.

Point 6: How did you control the confounders and bias in your statistical analysis?

Answer 6:

In the present study, which is of a descriptive nature, there was no need to control for confounder or bias on the statistical analysis performed.  

Point 7. Try to make your presentation of your findings well-structured.

Answer 7:

In the qualitative part of the study (interviews and focus group) we followed the tradition from the qualitative research, were excerpts from interviews are an important part of the results section, a kind of “proof” for discussion section. In accordance with your comment we have now shortened some sections. See page 12 lines 439-443, 450-451 and page 13 lines 469-472 and 499-500. We hope that this change has make the findings section more readable. Also, this section is structured in accordance with the most prevalent CFIR domains which we now clarified on page 11, line 407.

Looking forward to your next version! Good luck!

Reviewer 3 Report

This is an interesting study, but it came across as quite specialized and detailed. I wonder if there are some means of framing it for wider audiences. (Was this kind of information contained in the masked reference article cited?) Having worked in healthcare in the U.S. I thought of the many interventions we used over the years and how this may relate to those, for example, despite our different models.

Another audience (besides the targeted one) could be researchers and practitioners interested in process evaluations beyond this particular context of stress reduction in healthcare--is there anything at all they could glean from this study? I am considering change management interventions as broad and varied examples. Might this study confirm or dispute anything there?

Or even suggest any nuance to broader program evaluation audience that is less familiar with process evaluations? 

As a qualitative researcher, I felt that the authors were a bit too apologetic over the "subjectivity" of those findings, given the weakness of claims traditionally associated with such a method. "Intersubjectivity" (among participants, and extending to the researchers) and complementing the quantitative methods seem like adequate standards. 

I liked the tables and diagrams.

33" Organizational interventions are important for the primary prevention of stress and stress34 related mental ill-health because they target known environmental risk factors for mental ill-health
35 such as high job demands, low job control and low social support [1]. However, to date there is
36 limited research evidence that organizational-level interventions are effective in preventing stress
37 and mental ill-health, the only exception being interventions that include a change of work schedules,
38 i.e. shorter or interrupted work schedules [2]"

This was a little confusing--if interventions are "important," it is then surprising that there is "limited evidence" of their effectiveness. Maybe restate more specifically--are they "common"? Do they "show promise" or "have potential"? Could these two thoughts --importance and limitations--be more closely connected and combined?

I found a few typos.

Author Response

Reply to Comments and Suggestions for Authors

This is an interesting study, but it came across as quite specialized and detailed. I wonder if there are some means of framing it for wider audiences. (Was this kind of information contained in the masked reference article cited?)

Answer 1: yes, this kind of information is contained in the masked reference.

Having worked in healthcare in the U.S. I thought of the many interventions we used over the years and how this may relate to those, for example, despite our different models.

Another audience (besides the targeted one) could be researchers and practitioners interested in process evaluations beyond this particular context of stress reduction in healthcare--is there anything at all they could glean from this study? I am considering change management interventions as broad and varied examples. Might this study confirm or dispute anything there?

Or even suggest any nuance to broader program evaluation audience that is less familiar with process evaluations? 

Answer 2

Thank you for your time and effort. We understand your point.

We have now tried to make some points at page 2, lines 56-63 and 68-73 and also added some theoretical perspectives that may make it more interesting for other researchers, please see page 2, lines 75-94, page 3, lines 95-97 and page 5, lines 132-140.

In the discussion section we elaborated our discussion on page 15, lines 564-573.

We also added an important piece of information from the RCT on page 4, lines 123-125 that we followed up in the discussion on page 15, lines 584-588.

In conclusion section (page 18), we added lines 722-726.

As a qualitative researcher, I felt that the authors were a bit too apologetic over the "subjectivity" of those findings, given the weakness of claims traditionally associated with such a method. "Intersubjectivity" (among participants and extending to the researchers) and complementing the quantitative methods seem like adequate standards. 

Answer 3:

Thank you for your comment. We have now removed the paragraph on page 17, lines 700-707.

I liked the tables and diagrams.

Thank you

33" Organizational interventions are important for the primary prevention of stress and stress34 related mental ill-health because they target known environmental risk factors for mental ill-health
35 such as high job demands, low job control and low social support [1]. However, to date there is
36 limited research evidence that organizational-level interventions are effective in preventing stress
37 and mental ill-health, the only exception being interventions that include a change of work schedules,
38 i.e. shorter or interrupted work schedules [2]"

This was a little confusing--if interventions are "important," it is then surprising that there is "limited evidence" of their effectiveness. Maybe restate more specifically--are they "common"? Do they "show promise" or "have potential"? Could these two thoughts --importance and limitations--be more closely connected and combined?

Answer 4:

We have now changed this paragraph somewhat, please see page 1, lines 36-43

I found a few typos.

Answer 5: Thank you, we have tried to find them

Reviewer 4 Report

This study focus on a mixed method research study to evaluate the implementation process and fidelity to the intervention guidelines, intervention (Productivity Measurement and Enhancement System or ProMES) on employee stress. It examines the influence of contextual factors (hindrances and facilitators) and try to explore participants experience of working with ProMES. Therefore the research team use the UK Medical Research Council (MRC) guidance to guide the process evaluation. Some obstacles are decribed in the text. 1. This are difficulties in obtaining statistical productivity data from 
the central administration office because of less implementation 
and regular feedback meetings. 2. Further there was a leck in staff to do the research.

So it can be pointed out, that it is beneficial giving a detailed examination of access to necessary organizational data before implementing ProMES and giving a better introduction and program training for new employees.

In front of the qualitative and quantitative data collection I can reccommend a much better declaration of the evaluation system. That means concretely: What kind of mix method system is used? That could bring much more clearness about the final data and the research results. In the chapter „constraints“ it is a good way to make sure, where are strength and weaknesses of this study.   

Author Response

Reply to Comments and Suggestions for Authors

This study focus on a mixed method research study to evaluate the implementation process and fidelity to the intervention guidelines, intervention (Productivity Measurement and Enhancement System or ProMES) on employee stress. It examines the influence of contextual factors (hindrances and facilitators) and try to explore participants experience of working with ProMES. Therefore the research team use the UK Medical Research Council (MRC) guidance to guide the process evaluation. Some obstacles are decribed in the text. 1. This are difficulties in obtaining statistical productivity data from 
the central administration office because of less implementation 
and regular feedback meetings. 2. Further there was a leck in staff to do the research.

So it can be pointed out, that it is beneficial giving a detailed examination of access to necessary organizational data before implementing ProMES and giving a better introduction and program training for new employees.

Point 1:In front of the qualitative and quantitative data collection I can reccommend a much better declaration of the evaluation system. That means concretely: What kind of mix method system is used? That could bring much more clearness about the final data and the research results.

Thank you for your engagement in the review of our manuscript.

I apologize but I am not sure if I understood you properly. We used convergent parallel design (page 6, lines 184-186). However, we changed the wording in the paragraph on page 6, lines 181-183, hoping that is what you meant.

In the chapter „constraints“ it is a good way to make sure, where are strength and weaknesses of this study.  

We have some strengths and weaknesses described under “methodological considerations”. We have however in this version deleted some text as recommended by another IJERPH reviewer.

Round 2

Reviewer 2 Report

Congratulation!

Please add the future directions for further research in your implications section.

Author Response

Congratulation!

Thank You!

Point 1: Please add the future directions for further research in your implications section.

Answer 1: 

Thank you for a very relevant suggestion. We have now added following (page 17, lines 675-683):

4.2. Implications for future research

A study with a larger sample could shed more light on the effects of the intervention. In addition, there is a need to explore how to protect and support already vulnerable individuals during the implementation of any organizational intervention or change. One way to facilitate the above (as well as to facilitate the implementation of interventions in general) is to invite organizations much earlier in the research process, and establish practice based research networks, as recently described in a Swedish report on a practice based research network approach [53].

Round 3

Reviewer 2 Report

Congratulations!